# Parameter-Efficient Language Model Tuning with Active Learning in Low-Resource Settings

**Josip Jukić**    **Jan Šnajder**

TakeLab

Faculty of Electrical Engineering and Computing, University of Zagreb, Croatia

`{josip.jukic, jan.snajder}@fer.hr`

## Abstract

Pre-trained language models (PLMs) have ignited a surge in demand for effective fine-tuning techniques, particularly in *low-resource* domains and languages. Active learning (AL), a set of algorithms designed to decrease labeling costs by minimizing label complexity, has shown promise in confronting the labeling bottleneck. In parallel, adapter modules designed for parameter-efficient fine-tuning (PEFT) have demonstrated notable potential in low-resource settings. However, the interplay between AL and adapter-based PEFT remains unexplored. We present an empirical study of PEFT behavior with AL in low-resource settings for text classification tasks. Our findings affirm the superiority of PEFT over full-fine tuning (FFT) in low-resource settings and demonstrate that this advantage persists in AL setups. We further examine the properties of PEFT and FFT through the lens of forgetting dynamics and instance-level representations, where we find that PEFT yields more stable representations of early and middle layers compared to FFT. Our research underscores the synergistic potential of AL and PEFT in low-resource settings, paving the way for advancements in efficient and effective fine-tuning.[1]

## 1  Introduction

Pre-trained language models (PLMs) have quickly become a staple in the field of natural language processing. With the growing demand for data for training these models, developing efficient fine-tuning methods has become critical. This is particularly relevant for many domains and languages where obtaining large amounts of labeled training data is difficult or downright impossible. In such *low-resource settings*, it becomes essential to effectively leverage and adapt PLMs while minimizing the need for extensive labeled data.

Data labeling is notoriously time-consuming and expensive, often hindering the development of sizable labeled datasets required for training high-performance models. *Active learning* (AL) (Cohn et al., 1996; Settles, 2009) has emerged as a potential solution to this challenge. In contrast to *passive learning*, in which the training set is sampled at random, AL encompasses a unique family of machine learning algorithms specifically designed to reduce labeling costs by reducing *label complexity*, i.e., the number of labels required by an *acquisition model* to achieve a certain level of performance (Dasgupta, 2011). With the advent of PLMs, AL research has pivoted towards investigating training regimes for PLMs, such as task-adaptive pre-training (TAPT; Gururangan et al., 2020), that could be combined with AL to further reduce the label complexity.

While AL aims at directly minimizing the label complexity of learning, training efficiency can also be improved by reducing the *parameter complexity* of the model. This becomes more important as PLMs grow larger, and fine-tuning becomes increasingly challenging due to the sheer number of parameters involved. To address this issue, adapters (Houlsby et al., 2019) have been introduced as compact modules that can be incorporated between the layers of PLMs. Adapters enable considerable parameter-sharing, facilitating parameter-efficient fine-tuning (PEFT) through modular learning (Pfeiffer et al., 2023). In this process, only the parameters of the adapters are updated during the tuning for a specific downstream task. Recent research (He et al., 2021; Li and Liang, 2021; Karimi Mahabadi et al., 2021) has revealed that some PEFT methods outperform full fine-tuning (FFT) in low-resource settings, potentially due to better stability and a decreased risk of overfitting. In contrast, FFT has been shown to exhibit instability in scenarios with limited data.

Despite the promising results demonstrated by PEFT methods in low-resource settings, there is a

---

[1] Our code is available at https://github.com/josipjukic/adapter-al

striking gap in research on parameter-efficient training with respect to how PEFT interacts with AL. Given that the majority of real-world AL scenarios involve a restricted amount of data, PEFT methods emerge as strong candidates for AL acquisition models. However, there has been no exploration of AL in conjunction with adapters. Investigating this uncharted territory can further advance our understanding of AL and reveal novel strategies for optimizing performance in low-resource settings.

In this paper, we present an empirical study on the behavior of PEFT in low-resource settings for text classification tasks. We analyze PEFT with and without AL and compare it against FFT. While our results confirm that PEFT exhibits superior performance in low-resource setups compared to FFT, we show that the improved performance with PEFT extends to AL scenarios in terms of performance gains over passive learning. Furthermore, we analyze the efficacy of TAPT in conjunction with AL and PEFT. We find that TAPT is beneficial in AL scenarios for both PEFT and fully fine-tuned models, thus representing a viable technique for improving performance in low-resource settings. Finally, aiming to illuminate why PEFT and TAPT improve AL performance in low-resource settings, we analyze the properties of PEFT and FFT via *forgetting dynamics* (Toneva et al., 2019) and PLMs' instance-level representations. We find that AL methods choose fewer *unforgettable* and more *moderately forgettable* examples when combined with PEFT and TAPT, where forgetfulness indicates the model's tendency to learn and forget the gold label of a particular instance. Compared to FFT, we observe that PEFT yields representations in the **early** and **middle** layers of a model that are more similar to the representations of the base PLM. We hypothesize that this property mitigates the issue of forgetting the knowledge obtained during pretraining when fine-tuning for downstream tasks.

In summary, we show that in AL low-resource settings for text classification, (1) PEFT yields greater performance improvements compared to FFT and (2) TAPT enhances the overall classification performance of adapters and is well-suited for AL scenarios. We also show that (3) AL methods choose fewer unforgettable and more moderately forgettable examples with PEFT and that (4) PEFT produces instance-level representations of early and middle layers that are more similar to the base PLM than FFT. Our results uncover the intrica-

cies of positive interactions between AL, PEFT, and TAPT, providing empirical justification for their combined use in low-resource settings.

## 2 Related Work

Our research involves combining AL with PLMs and investigating the use of PEFT techniques within the confines of low-resource settings.

**AL with PLMs.** Until recently, the conventional approach for integrating PLMs with AL involved performing full fine-tuning with a fixed number of training epochs and training the model from scratch in each AL step (Ein-Dor et al., 2020; Margatina et al., 2021; Shelmanov et al., 2021; Karamcheti et al., 2021; Schröder et al., 2022). However, studies by Mosbach et al. (2021) and Zhang et al. (2021) revealed that fine-tuning in low-resource setups is prone to instability, particularly when training for only a few epochs. This instability, often sensitive to weight initialization and data ordering (Dodge et al., 2020), presents a significant challenge for AL, which frequently operates in low-resource settings. Recent research has looked into the impact of PLM training regimes on AL performance (Grießhaber et al., 2020; Yuan et al., 2020; Yu et al., 2022), suggesting that the choice of training regime is more critical than the choice of the AL method. Notably, TAPT has proven particularly effective in enhancing AL performance (Margatina et al., 2022; Jukić and Šnajder, 2023).

**Adapters in low-resource settings.** Research on adapters in low-resource settings has primarily focused on areas such as cross-lingual transfer for low-resource languages (Ansell et al., 2021; Lee et al., 2022; Parović et al., 2022), where the emphasis lies on exploring diverse methods of fusing adapters. In monolingual settings with scarce data, adapters have been found to outperform full fine-tuning (Li and Liang, 2021; Mao et al., 2022). A study by He et al. (2021) demonstrated that adapter-based tuning exhibits enhanced stability and generalization capabilities by virtue of being less sensitive to learning rates than traditional fine-tuning methods. While incorporating task adaptation techniques, such as TAPT, has been shown to match or even improve performance over FFT in low-resource setups, Kim et al. (2021) noted an interesting caveat: the benefits of integrating TAPT with adapters tend to taper off as the amount of data increases.

Despite the established effectiveness of adapters in setups with limited resources, their integration into AL frameworks — which frequently face analogous resource constraints — remains an untapped area of research. This gap is particularly notable given that AL's iterative learning process could significantly benefit from adapters' parameter efficiency and transferability, especially in scenarios where data scarcity or labeling costs are primary concerns.

## 3 Preliminaries

We now describe the experimental setup, providing details on the datasets as well as the PEFT and AL methods used in our study.

### 3.1 Datasets

We employ four single-text classification tasks commonly used for AL evaluation: (1) the subjectivity dataset (**SUBJ**; Pang and Lee, 2004), designed to assess the subjectivity of a given text; (2) the question type classification dataset (**TREC**; Li and Roth, 2002), designed for categorizing questions according to their types; (3) the Stanford Sentiment Treebank (**SST**; Socher et al., 2013), which focuses on sentiment analysis; (4) AG's news classification dataset (**AGN**; Zhang et al., 2015), which classifies news articles into different categories. We provide the dataset statistics in the appendix for further reference (cf. Appendix Table 3).

### 3.2 PEFT methods

We consider four prototypical PEFT techniques:

**Adapter** incorporates trainable bottleneck layers after both the multi-head attention and feedforward block in each Transformer layer (Houlsby et al., 2019);

**Prefix-tuning** adds new parameters in the multi-head attention blocks within each Transformer layer (Li and Liang, 2021);

**LoRA** (**Lo**w-**r**ank **a**daptation) represents an additive method that incorporates trainable low-rank decomposition matrices into the layers of a pre-trained model (Hu et al., 2022);

**UniPELT** combines multiple PEFT approaches, namely LoRA, Prefix-tuning, and Adapter, in a single unified setup (Mao et al., 2022). Each constituent is a submodule, and UniPELT employs gating mechanisms to activate them effectively.

All of the above PEFT methods fall under the category of lightweight fine-tuning. While prefix-tuning does not technically qualify as an adapter, He et al. (2022) demonstrated that it shares formal similarities with adapters, with prefix-tuning performing weighted addition and an adapter employing unweighted addition. We refer to all four considered methods as adapters for terminological simplicity. We use BERT (Devlin et al., 2019) as the base PLM for every adapter. Additionally, we adhere to the hyperparameter settings for each adapter as recommended in the respective papers that introduced them (cf. Appendix A.2 for details).

### 3.3 AL methods

Our study considers five sampling strategies, including **random selection** (**RND**) as a passive learning baseline. The other four strategies are AL methods originating from different families, chosen for their robustness (ability to perform well across various tasks) and widespread usage in the field:

**Maximum entropy** (**ENT**; Lewis and Gale, 1994) comes from the family of *uncertainty* strategies. The method queries instances where the model is least certain based on the maximum entropy criterion of the prediction output;

**Monte Carlo dropout** (**MC**; Gal and Ghahramani, 2016) resembles ENT but utilizes the stochasticity of forward passes with dropout layers (Srivastava et al., 2014) to estimate the entropy for a given instance;

**Core-set** (**CS**; Sener and Savarese, 2018) encourages instance diversity by using the learned representations of the acquisition model. This method aims to minimize the distance between an example in the unlabeled set and its closest counterpart in the labeled subset;

**Discriminative active learning** (**DAL**; Gissin and Shalev-Shwartz, 2019) frames AL as a binary classification of instances into those that are labeled and those that are not, with the objective of making the labeled and unlabeled sets indistinguishable.

### 3.4 Experimental setup

In AL runs, we select 50 new examples in each step of each AL experiment, using 100 examples for the warm start (randomly sampled labeled data

to initiate the model). To probe different PEFT approaches with and without AL in low-resource settings, we establish a labeling budget limit of $1,000$ instances. To sidestep the need for a validation set in our experiments, which is typically unavailable in real-world AL scenarios, we adopt the Besov early stopping (Jukić and Šnajder, 2023). This method utilizes the smoothness of Transformer layers to decide at which epoch to stop training.

In the case of TAPT, we pre-train the base model on a masked language modeling task using unlabeled training data. For adapters, we only update the injected parameters while keeping the remaining parameters of the base model frozen. This approach aligns with the primary function of adapters, which is to utilize a common base model across diverse tasks. For every setting, we perform five runs using different random seeds. We report the average $F_1$ score at each sampling step (with and without AL for FFT and PEFT) to show the corresponding learning curve averaged over five runs. We provide details on training and hyperparameters in Appendix A.5.

## 3.5 Evaluation

To evaluate the overall performance of an AL method, we employ the area under the performance curve (AUC). In each individual AL step with a specific quantity of labeled examples, we measure the classification performance in terms of the $F_1$ score. The overall AUC is calculated using the $F_1$ scores obtained at each step. We advocate for using AUC alongside the AL curves, as AUC serves as a suitable approximation of AL feasibility through a summary numeric score, as recommended in recent AL literature (Schröder et al., 2022; Jukić and Šnajder, 2023).

As our experiments involve different training regimes, we compare each AL sampling strategy $S_{\mathrm{AL}}$ to passive learning $S_{\mathrm{PL}}$ within the same training regime to isolate the effects of AL. The primary objective of AL is to improve label efficiency over passive learning. To test whether AL is successful, we calculate the **r**elative **i**mprovement over **p**assive **l**earning (RIPL), which we define as follows:

$$\mathrm{RIPL}(S_{\mathrm{AL}}, S_{\mathrm{PL}}) = \frac{\mathrm{AUC}(S_{\mathrm{AL}}) - \mathrm{AUC}(S_{\mathrm{PL}})}{1 - \mathrm{AUC}(S_{\mathrm{PL}})}$$

Intuitively, RIPL estimates the proportion of maximum possible improvement achievable by a given AL method compared to the passive learning baseline. A score of 1 indicates the maximum theoret-

ical improvement, which would be tantamount to attaining an $F_1$ score of 1 in the initial sampling step and sustaining that score throughout all steps. Conversely, a negative score indicates that the AL method performs worse than passive learning.

## 4 Experiments

In this section, we first examine the performance of PEFT methods in comparison to FFT with passive learning and then proceed to analyze the application of PEFT in AL settings.

### 4.1 PEFT vs. FFT

Previous research on the use of adapters in low-resource settings (Li and Liang, 2021; Mao et al., 2022; He et al., 2021) has demonstrated that adapters perform comparable to, and sometimes even better than FFT. However, these findings were based on comparing FFT to a single adapter variant on a full dataset or evaluating the performance at only a few discrete points.

In the first part of our experiments, we build upon these findings by conducting a more nuanced analysis. We generate detailed learning curves that facilitate the comparison of multiple adapters with FFT under the passive learning setup. Our comparison, summarized by the AUC metric in Table 1, reveals that UniPELT and Prefix-tuning consistently outperform FFT with a significant difference across all datasets used in our study. Conversely, the performance of Adapter and LoRA is mostly comparable to FFT, although there are cases where they either outperform or underperform FFT. In cases in which Adapter and LoRA perform better than FFT with significant differences, the degree of improvement is smaller than what is observed with UniPELT and Prefix-tuning.

Next, we look into how the models' performance changes as the training set increases. To that end, we show the corresponding learning curves for adapters and FFT in Figure 1. The performance disparities between adapters and FFT become more apparent under conditions of extreme data scarcity (100–300 labeled instances). Notably, the greatest differences in performance occur at the initial step (only 100 labels). This highlights the promise of adapter-based methods in low-resource settings, particularly for Prefix-tuning and UniPELT.

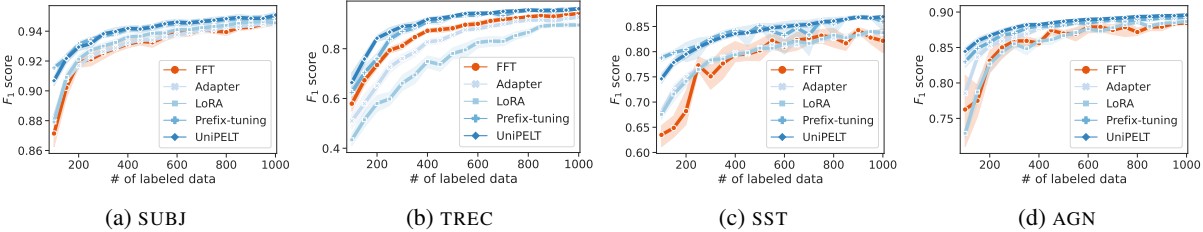

<table>
<tr><td></td><td>(a) SUBJ</td><td>(b) TREC</td><td>(c) SST</td><td>(d) AGN</td></tr>
</table>

Figure 1: Learning curves under the passive learning setup with different PEFT methods and FFT. The results are averaged over five runs. The shaded bands denote the standard deviation. Best viewed on a computer screen.

|          |               | SUBJ              | TREC              | SST               | AGN               |
|----------|---------------|-------------------|-------------------|-------------------|-------------------|
| adapters | Adapter       | .926              | .804              | .800$^\dagger$     | .871$^\dagger$     |
|          | LoRA          | .929              | .750$^\dagger$     | .798$^\dagger$     | .860              |
|          | Prefix-tuning | **.936**$^\dagger$ | .847$^\dagger$    | **.847**$^\dagger$ | **.875**$^\dagger$ |
|          | UniPELT       | .934$^\dagger$    | **.877**$^\dagger$ | .836$^\dagger$    | **.875**$^\dagger$ |
|          | FFT           | .928              | .810              | .787              | .860              |

Table 1: The performance of adapters and FFT in a passive learning setup in terms of the AUC metric (based on $F_1$ score) averaged over five runs. Numbers in **bold** represent the best-performing variant for a particular dataset. The "†" symbol indicates when the mean AUC of an adapter is significantly different from the corresponding mean AUC of FFT ($p < .05$ using a two-sided Man-Whitney U test adjusted for family-wise error rate with the Holm-Bonferroni method).

## 4.2 PEFT with AL

Motivated by our initial findings on using PEFT under the passive learning setup, where PEFT exhibited promising properties in low-resource settings, we further explore the behavior of adapters in AL scenarios. We evaluate individual PEFT methods in AL scenarios with and without using TAPT in terms of gains over random sampling (passive learning) using the RIPL metric described in Section 3.5. Table 2 shows the results for different combinations of AL methods and adapters, evaluated through the RIPL metric. We complement these results with absolute values in terms of AUC (cf. Appendix Table 5). For FFT without TAPT, DAL achieved the highest RIPL score on two datasets, while CS and MC topped the chart on one dataset each. When we incorporated TAPT, ENT yielded the best results on three out of four datasets, with CS leading on one. Looking at adapters, the most successful AL methods without TAPT vary, depending on the specific adapter and dataset in question. Interestingly, when TAPT is applied, the best results for all adapters are obtained either by ENT or MC. We speculate this could be attributed to solid compatibility between

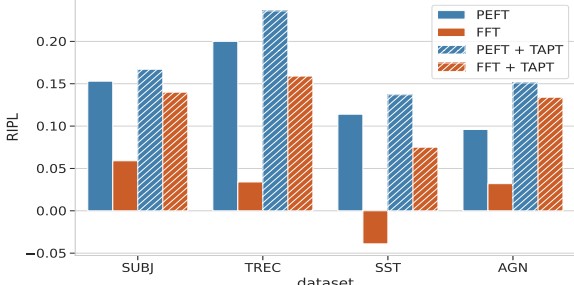

Figure 2: Comparison of best-performing adapters and FFT from Table 2 and their corresponding version with TAPT applied.

entropy-based methods and TAPT when adapters are employed.

Furthermore, we observe that without TAPT, adapters achieve larger gains over FFT. However, when TAPT is applied, FFT becomes comparable to PEFT, although Prefix-tuning and UniPELT still yield the greatest improvements, depending on the dataset and AL method used. In Figure 2, we select the adapters that achieved the best improvement according to Table 2 without TAPT and show their RIPL value compared against FFT as well as their corresponding version when TAPT is applied. We conjecture that TAPT reduces the performance gap between adapters and FFT by inducing FFT to emulate PEFT in aspects such as training dynamics and representation space — a hypothesis we explore in more detail in Section 5.

We further investigate the behavior of adapters with AL throughout the individual steps. Figure 3 shows the learning curves for corresponding adapter models with and without applying TAPT. Due to space constraints, we show the learning curves only for the SUBJ dataset, as similar trends occur for other datasets. Without TAPT, the performance of adapters is largely independent of the specific AL method used, where Prefix-tuning and UniPELT consistently outperform Adapter and LoRA across all AL steps. With TAPT, the differ-

| | | without TAPT | | | | with TAPT | | | |
| --- | --- | --- | --- | --- | --- | --- | --- | --- | --- |
| | | ENT | MC | CS | DAL | ENT | MC | CS | DAL |
| SUBJ | FFT | .050 | .059 | .061 | .077 | .140 | .140 | .142 | .126 |
| | Adapter | .112 | .102 | .100 | .092 | .137 | .151 | .111 | .067 |
| | LoRA | .127 | .115 | .091 | .081 | .165 | .160 | .122 | .100 |
| | Prefix-tuning | .095 | .110 | .106 | .111 | **.186** | .181 | .170 | .151 |
| | UniPELT | .129 | **.153** | .131 | .128 | .159 | .167 | .163 | .157 |
| TREC | FFT | .011 | .022 | .038 | .034 | .162 | .180 | .141 | .159 |
| | Adapter | .027 | .069 | .137 | .084 | .124 | .146 | .079 | .154 |
| | LoRA | .098 | .065 | .048 | .007 | .254 | .237 | .243 | .074 |
| | Prefix-tuning | .093 | .105 | .068 | .093 | .246 | .227 | .205 | .241 |
| | UniPELT | .138 | .165 | .082 | **.200** | .302 | **.334** | .276 | .236 |
| SST | FFT | .002 | .011 | −.039 | .004 | .080 | .079 | .075 | .070 |
| | Adapter | .015 | .048 | .025 | .002 | .035 | .034 | .028 | .008 |
| | LoRA | .001 | .007 | .064 | .031 | .036 | .022 | .032 | .014 |
| | Prefix-tuning | .049 | .060 | **.114** | .031 | **.152** | .143 | .137 | .126 |
| | UniPELT | .037 | .043 | .040 | .008 | .082 | .101 | .083 | .080 |
| AGN | FFT | .014 | .032 | .007 | .092 | .134 | .021 | .089 | .017 |
| | Adapter | .074 | .046 | .015 | .062 | .115 | .089 | .077 | .080 |
| | LoRA | .020 | .025 | .067 | .016 | .028 | .102 | .071 | .023 |
| | Prefix-tuning | .054 | .023 | .040 | .033 | .035 | .143 | .098 | .092 |
| | UniPELT | .074 | **.096** | .089 | .095 | **.185** | .151 | .112 | .081 |

Table 2: Improvement over passive learning in terms of the RIPL metric for four AL methods considered (ENT, MC, CS, and DAL) and for all combinations of adapters and datasets considered, shown separately without TAPT and with TAPT. Positive values indicate improvement over passive learning, while negative values indicate performance drops compared to passive learning. Values in **bold** denote the best result for a particular dataset across different adapters and AL methods within the same regime (with or without TAPT).

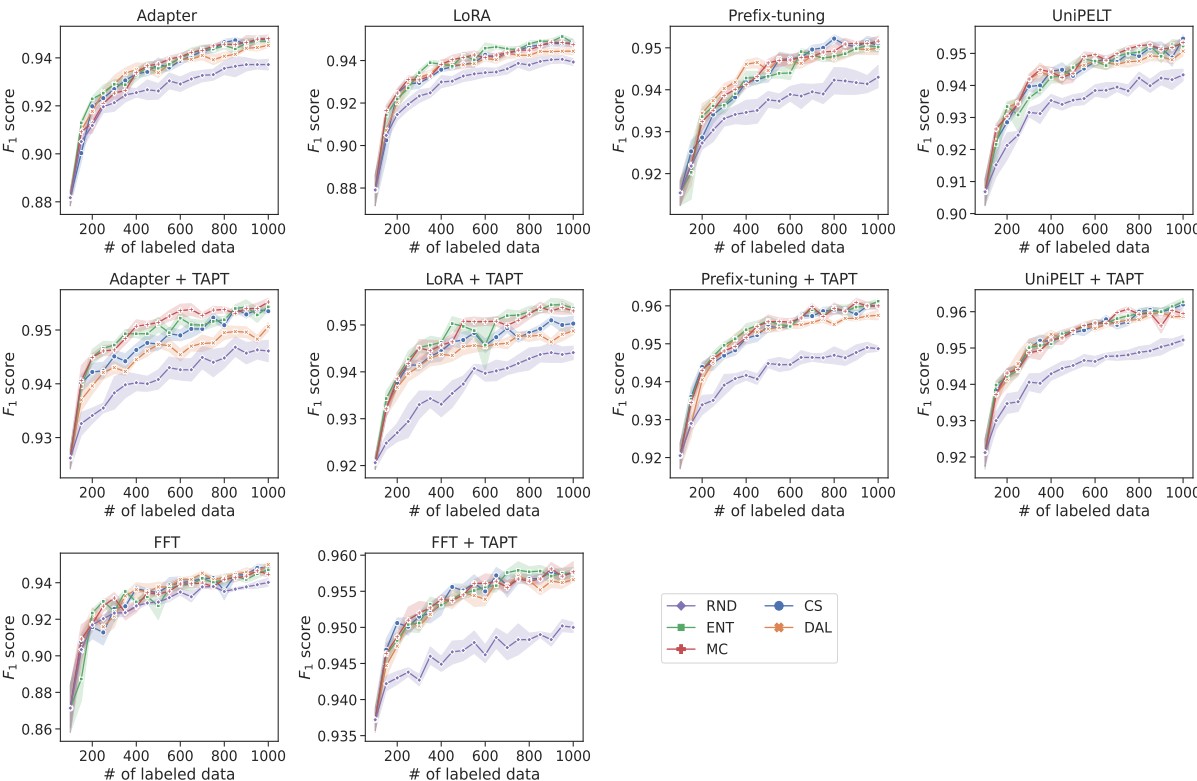

Figure 3: AL learning curves compared with random sampling on the SUBJ dataset. The first and the second rows show learning curves for adapters without and with TAPT, respectively. The third row shows learning curves for FFT, without and with TAPT. The results are averaged over five runs, and the shaded bands denote the standard deviation. Best viewed on a computer screen.

ences between AL and random sampling are more pronounced starting from the early steps, typically already with 200 instances. In contrast, the gap becomes more apparent only with 500 or more instances when TAPT is not employed.

# 5 Analysis

In Section 4, we have demonstrated that PEFT exhibits larger gains than FFT when combined with AL in low-resource settings, which is also accompanied by superior performance with passive leaning. To better understand why PEFT displays superior behavior with limited data, we now examine two specific properties of adapters and FFT models. First, we analyze the influence of TAPT on the forgetting dynamics during training. We continue with example-level representation analysis, where we investigate the representation similarity of PEFT and FFT to their respective base models.

## 5.1 Forgetting dynamics

We employ forgetting dynamics to compare PEFT and FFT's learning stability and their impact on AL data selection. The underlying hypothesis is that having fewer forgetting events in adapters would indicate a more stable and effective learning process. In utilizing forgetting dynamics, we draw upon the study by Toneva et al. (2019), focusing on the occurrence of *forgetting events* — cases where a specific training example transitions from correct to incorrect classification over the course of multiple learning epochs. More specifically, we divide the instances into three categories: (1) **unforgettable** instances, i.e., the ones that have never experienced a forgetting event during training, (2) instances that have encountered one or two forgetting events, labeled as **moderately** forgettable, and (3) instances subjected to three or more forgetting events, referred to as **highly** forgettable instances. As pointed out in the original study, moderately forgettable, *ambiguous* instances are more valuable for the learning model than unforgettable, *easy* instances. However, it is worth noting that AL is often hindered by too *hard* or *impossible-to-learn* examples (Karamcheti et al., 2021), which roughly correspond to the highly forgettable examples.

Figure 4 shows the distribution of instances across the three categories of forgetting events for SUBJ and TREC datasets. We focus on these two datasets as examples of a simple binary classification task and a more complex multi-class classi-

fication task, respectively. Specifically, we compare RND with MC, which achieves consistent performance improvements across all datasets. Our findings suggest that FFT tends to select a higher number of unforgettable instances and fewer moderately forgettable instances when compared to adapters. Interestingly, the adapters that perform best — Prefix-tuning and UniPELT — appear to favor moderately forgettable instances. However, when TAPT is applied, the discrepancies in forgetting profiles between FFT and the top two adapters, Prefix-tuning and UniPELT, seem to diminish. In contrast, TAPT amplifies the differences between FFT and the other two adapters, LoRA and Adapter, which typically show smaller improvements than Prefix-tuning and UniPELT. Given their superior AL performance, we hypothesize that the forgetting profiles of Prefix-tuning and UniPELT are more favorable compared to other adapters. Moreover, FFT with TAPT approaches the performance of the superior adapters and simultaneously develops a forgetting profile similar to theirs.

## 5.2 Representation analysis

To bolster our findings, we explore the representations of adapters and FFT models. As suggested in previous research (He et al., 2021; Li and Liang, 2021; Mao et al., 2022), adapters often display greater stability in terms of loss, especially in scenarios with limited resources. Our aim is to examine the stability of their representations and their relationship with overall AL performance.

We draw inspiration from research by Stephenson et al. (2021) and Baldock et al. (2021), which suggests that different layers of networks specialize in different features — earlier layers tend to acquire more generalized knowledge, while the deeper layers are more focused on task-specific information. This leads us to a layerwise examination of similarity. To analyze the effect of PEFT and FFT on AL selection with respect to their layerwise similarity to the base model, we utilize centered kernel alignment (CKA) as a similarity measure between two sets of representations (Kornblith et al., 2019). It has been shown that PEFT methods result in representations closer to the base model at the token level (He et al., 2021). We extend the analysis to example-level representation to explore the behavior of models with AL. We opt for CKA as it is designed to be invariant to invertible linear transformation and still can measure meaningful similari-

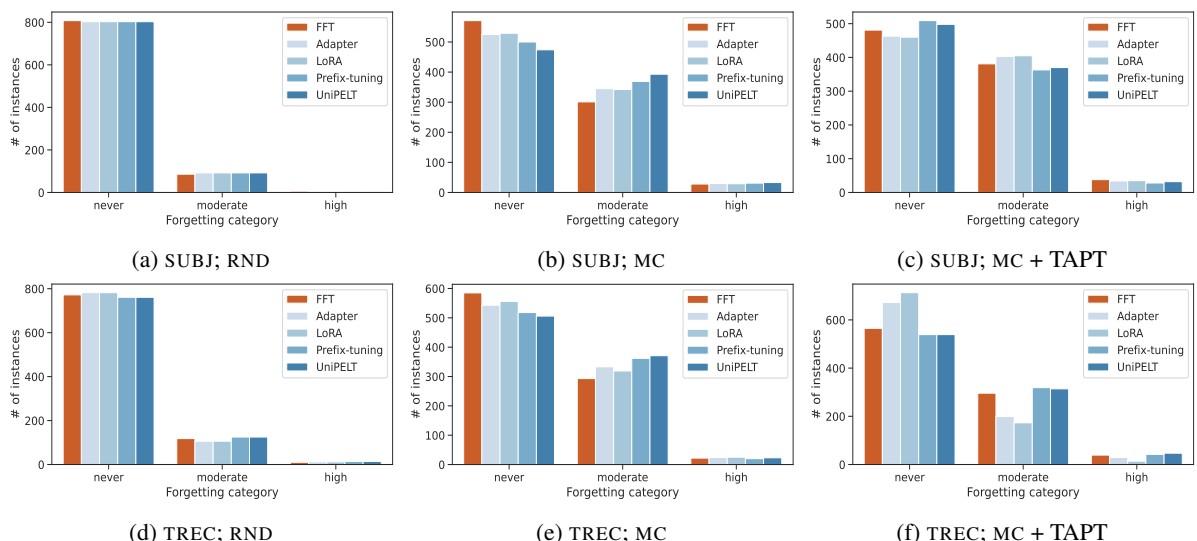

(a) SUBJ; RND       (b) SUBJ; MC       (c) SUBJ; MC + TAPT

(d) TREC; RND       (e) TREC; MC       (f) TREC; MC + TAPT

Figure 4: Forgetting dynamics for random sampling (passive learning) and AL with MC without and with TAPT on SUBJ and TREC. The x-axis shows the number of instances in each of the forgetting categories: the "never" category representing **unforgettable** instances, **moderately** forgettable instances, and **highly** forgettable instances.

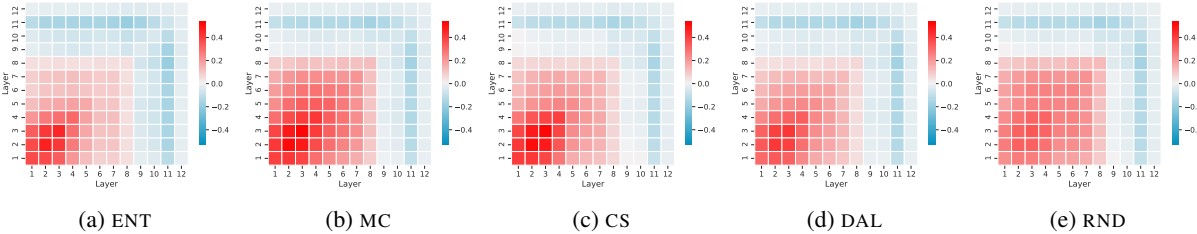

(a) ENT      (b) MC      (c) CS      (d) DAL      (e) RND

Figure 5: Layerwise difference in representation similarity for the UniPELT adapter and the FFT model on SUBJ. We observed similar patterns in other adapters and datasets we used (cf. Appendix B). Warm colors (positive values) illustrate layer pairs that demonstrate higher similarity to the base model with the adapter than with FFT. Conversely, cool colors (negative values) represent layer pairs that are more similar to the base model when using the FFT model. Best viewed on a computer screen.

ties between representations of higher dimensions than the number of data points. This stands in contrast to other metrics, which frequently falter when dealing with high-dimensional representations.

For a more direct comparison between PEFT and FFT, we analyze the differences between their respective similarities to their base models. Specifically, we compute the difference $\mathrm{CKA}(adapter, base) - \mathrm{CKA}(FFT, base)$ for a specific adapter or FFT and their base models. We hypothesize that superior PEFT performance with AL compared to FFT will be accompanied by a more similar early layer representation to the base model in PEFT. Figure 5 visualizes the layerwise difference in similarity between the base model and the adapter model and between the base model and the FFT model. We find that PEFT representations are more similar to the base model in the **early** and **middle** layers when compared to FFT. This

holds for all AL methods, with differences more pronounced than in passive learning. Specifically, up to the eighth layer, representations are much more similar in adapters than in FFT models. In the final four layers, the difference in CKA scores between the adapter and FFT model is close to zero. Interestingly, the penultimate layer is more similar in the FFT model with respect to the base model.

When fine-tuning on a downstream task, we believe that the increased stability of PEFT in earlier layers, relative to FFT, is instrumental in retaining the foundational knowledge from the PLM's pre-training phase. Conversely, PEFT exhibits more substantial transformations in the later, more task-specific layers. This ensures the preservation of essential pre-trained knowledge while allowing for task-relevant flexibility. We speculate that this strategic balance in PEFT influences its propensity to select moderately forgettable instances when

combined with AL, contributing to its enhanced performance over FFT. These instances are neither too trivial to provide no learning value, nor are they too complex to risk misinterpretation, thereby enhancing the effectiveness of learning.

# 6   Conclusion

Our study has shed light on the advantages of parameter-efficient fine-tuning (PEFT) in low-resource settings, confirming its superiority over full fine-tuning (FFT) methods. Importantly, we have demonstrated that the integration of PEFT with active learning (AL) can offer substantial performance gains compared to passive learning, even in settings where labeled data is scarce. Furthermore, we highlighted the potential of task-adaptive pre-training (TAPT) to improve model performance further when used in conjunction with both PEFT and AL. We found that AL methods, in combination with PEFT, tend to select fewer *unforgettable* instances and more *moderately forgettable* examples. We further found that PEFT maintains the integrity of **early** and **middle** layer representations similar to the base model. We conjecture that this property mitigates forgetting during downstream task fine-tuning. These insights inform us of a possible underpinning mechanism that contributes to PEFT's superior performance and stability in low-resource settings. Overall, our work highlights the potential of PEFT and AL and establishes a foundation for developing increasingly efficient and cost-effective approaches for training models in low-resource settings.

## Limitations

While our study advances the understanding of PEFT and AL's interaction in low-resource settings and uncovers intriguing insights about the forgetting dynamics during fine-tuning, it has a number of limitations.

To begin with, we have focused on text classification tasks, which are but one aspect of the wide range of potential applications for PLMs. Different tasks such as question answering, translation, or summarization might exhibit different behaviors under the same conditions. Consequently, the observed advantages of PEFT in the context of AL might not necessarily translate to other NLP tasks.

Next, our results are limited to the specific PLMs, AL strategies, and PEFT methods we have examined in this study. While we have attempted to be comprehensive in our experiments, the outcomes might vary with different models, strategies, or methods. For example, the effectiveness of AL combined with PEFT might differ if other AL strategies are employed. Similarly, different types of adapter architectures could potentially lead to different results.

Although we found that PEFT methods produce instance-level representations of early and middle layers more similar to the base PLM than FFT, a comprehensive understanding of how and why this similarity leads to increased stability and performance in low-resource settings is still lacking. Our hypothesis about the role of early and middle layer stability in mitigating the issue of forgetting the knowledge obtained during pre-training needs further substantiation.

Finally, it is important to acknowledge the complexity and multifaceted nature of forgetting dynamics. While our investigation provides valuable insights about the interaction of forgetting with PEFT and TAPT in AL scenarios, a deeper understanding of the mechanisms of forgetting in the context of large PLMs is needed. Particularly, it would be interesting to investigate whether the balance between unforgettable and moderately forgettable instances selected by the AL methods changes as the size of the model or the amount of available data changes.

Future work should aim to address these limitations and further explore the mechanisms behind the promising results obtained with the combination of PEFT and AL. This will contribute to a more comprehensive understanding of the interaction between AL and PLMs, and help refine strategies for efficient fine-tuning in low-resource settings.

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

| | TRAIN | VAL | TEST | TOTAL |
|---|---|---|---|---|
| SUBJ | 7,000 | 1,000 | 2,000 | 10,000 |
| SST | 6,647 | 868 | 1,425 | 8,940 |
| TREC | 4,881 | 452 | 500 | 5,833 |
| AGN | 20,000 | 7,600 | 7,600 | 35,200 |

Table 3: Dataset sizes by splits. Although we do not use a validation set (VAL) in our experiments, we report its size for completeness. For the AGN dataset, we performed uniform subsampling to ensure the computational feasibility of the experiments.

# A  Reproducibility

## A.1  Dataset statistics

The sizes of the datasets per split are provided in Table 3. Predominantly, the datasets encompass texts in English.

## A.2  Adapters

We use the implementation of adapters from AdapterHub (Pfeiffer et al., 2020).

**Adapter** We set reduction factor to 16 and use *swish* function as nonlinearity.

**LoRA** We include LoRA to the self-attention weights, intermediate, and output MLP weights of a model. We set the rank of the LoRA layer and the scaling factor $\alpha$ to 8.

**Prefix-tuning** We use *tanh* activation for Prefix-tuning, with prefix length set to 30 and bottleneck size of 512.

**UniPELT** We use Adapter, LoRA, and Prefix-tuning as components of UniPELT with the same hyperparameters as described for individual components. The only exception is that we set the prefix length for Prefix-tuning to 10 instead of 30.

## A.3  AL methods

**MC** In experiments, we use ten inference cycles to approximate the entropy of the output via Monte-Carlo dropout sampling.

**CS** We use the [CLS] token representation from the Transformer's penultimate layer. We follow the greedy method described in the original work (Sener and Savarese, 2018)

| | FFT | Adapter | LoRA | Prefix-tuning | UniPELT |
|---|---|---|---|---|---|
| SUBJ | 40.8 | 4.3 | 3.2 | 4.1 | 6.2 |
| TREC | 68.4 | 4.9 | 5.4 | 5.9 | 8.4 |
| SST | 43.9 | 5.1 | 5.0 | 4.9 | 7.6 |
| AGN | 72.1 | 7.3 | 6.1 | 4.4 | 9.3 |

Table 4: Experiment duration in minutes for all models across datasets. We report the average runtime over five different runs and five different sampling methods (five AL methods and random sampling).

## A.4  Preprocessing

We undertake a few pre-processing steps: convert all tokens to lowercase, eliminate non-alphanumeric tokens, and limit the token sequence to a maximum length of 200.

## A.5  Hyperparameters

We use a fixed learning rate of $2 \times 10^{-5}$ for FFT and $10^{-4}$ for adapters. Additionally, we set the gradient clipping to 1 during training. In our implementation of TAPT, we randomly mask 15% of tokens for both FFT models and adapters and train the model for 50 epochs with the learning rate set to $10^{-5}$.

## A.6  Computing infrastructure

We conducted our experiments on $4\times$ *AMD Ryzen Threadripper 3970X 32-Core Processors* and $4\times$ *NVIDIA GeForce RTX 3090* GPUs with 24GB of RAM. We used *PyTorch* version 1.9.0 and CUDA 11.4.

## A.7  Average runtime

We report the average runtime of experiments in Table 4.

# B  Additional Results

We report the results that were omitted from the main part of the paper due to space constraints. Table 5 shows AUC scores for different combinations of AL methods and adapters, complementing the relative improvement scores as AUC represents absolute scores for each configuration. In Figure 6, we display the difference in similarities of adapters and FFT compared to their base models on the remaining three datasets.

| | | without TAPT | | | | | with TAPT | | | | |
|---|---|---|---|---|---|---|---|---|---|---|---|
| | | RND | ENT | MC | CS | DAL | RND | ENT | MC | CS | DAL |
| SUBJ | FFT | .928 | .931 | .932 | .932 | .934 | .938 | .947 | .947 | .947 | .946 |
| | Adapter | .926 | .934 | .933 | .933 | .932 | .934 | .943 | .944 | .941 | .938 |
| | LoRA | .929 | .938 | .937 | .935 | .934 | .935 | .946 | .945 | .943 | .942 |
| | Prefix-tuning | .936 | .942 | .943 | .943 | .943 | .940 | .951 | .951 | .950 | .949 |
| | UniPELT | .934 | .943 | **.944** | .943 | .942 | .943 | .952 | **.953** | .952 | .952 |
| TREC | FFT | .810 | .812 | .814 | .817 | .816 | .818 | .847 | .851 | .844 | .847 |
| | Adapter | .804 | .809 | .818 | .831 | .820 | .820 | .842 | .846 | .834 | .848 |
| | LoRA | .750 | .775 | .766 | .762 | .752 | .764 | .824 | .820 | .821 | .781 |
| | Prefix-tuning | .847 | .861 | .863 | .857 | .861 | .862 | .896 | .893 | .890 | .895 |
| | UniPELT | .877 | .894 | .897 | .887 | **.902** | .896 | .927 | **.931** | .925 | .921 |
| SST | FFT | .787 | .787 | .789 | .779 | .788 | .792 | .809 | .808 | .808 | .807 |
| | Adapter | .800 | .803 | .810 | .805 | .801 | .812 | .819 | .818 | .817 | .814 |
| | LoRA | .798 | .798 | .799 | .811 | .804 | .806 | .813 | .810 | .812 | .809 |
| | Prefix-tuning | .847 | .854 | .856 | **.864** | .852 | .868 | **.888** | .887 | .886 | .885 |
| | UniPELT | .836 | .842 | .843 | .843 | .837 | .871 | .882 | .884 | .882 | .881 |
| AGN | FFT | .860 | .862 | .864 | .861 | .873 | .869 | .887 | .872 | .881 | .871 |
| | Adapter | .871 | .881 | .877 | .873 | .879 | .882 | .896 | .893 | .891 | .891 |
| | LoRA | .860 | .863 | .863 | .869 | .862 | .868 | .872 | .881 | .877 | .871 |
| | Prefix-tuning | .875 | .882 | .878 | .880 | .879 | .886 | .890 | .902 | .897 | .896 |
| | UniPELT | .875 | .884 | **.887** | .886 | **.887** | .887 | **.908** | .904 | .900 | .896 |

Table 5: AUC scores for AL methods with different adapters shown separately without TAPT and with TAPT. We include random sampling for comparison with AL methods. Values in **bold** denote the best result for a particular dataset within the same regime (with or without TAPT).

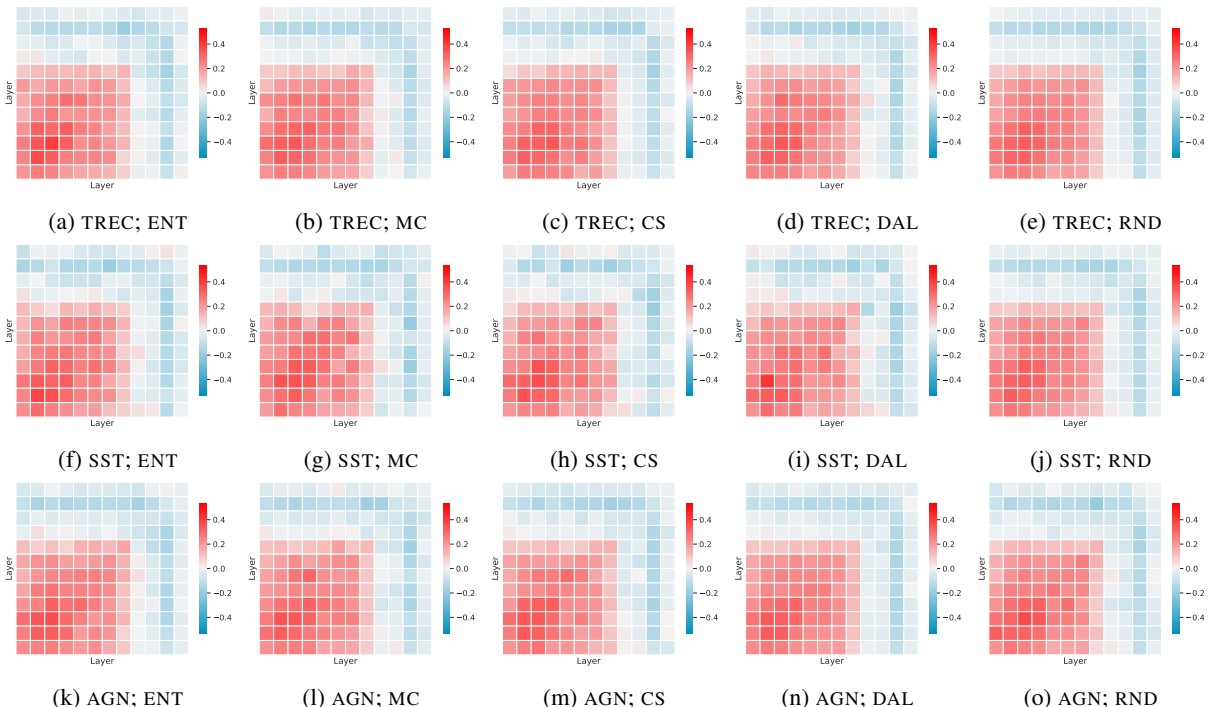

(a) TREC; ENT    (b) TREC; MC    (c) TREC; CS    (d) TREC; DAL    (e) TREC; RND

(f) SST; ENT    (g) SST; MC    (h) SST; CS    (i) SST; DAL    (j) SST; RND

(k) AGN; ENT    (l) AGN; MC    (m) AGN; CS    (n) AGN; DAL    (o) AGN; RND

Figure 6: Layerwise difference in representation similarity for the UniPELT adapter and the FFT model on TREC, SST, and AGN. The differences are computed as CKA(*adapter*, *base*) − CKA(*FFT*, *base*), where *base* is the corresponding pre-trained BERT model. Warm colors (positive values) illustrate layer pairs that demonstrate higher similarity to the base model with the adapter than with FFT. Conversely, cool colors (negative values) represent layer pairs that are more similar to the base model when using the FFT model. Best viewed on a computer screen.