# OpenReview forum: "Parameter-Efficient Language Model Tuning with Active Learning in Low-Resource Settings"
_EMNLP/2023/Conference — EMNLP 2023 Main_

### Official Review · Reviewer_qrFB · 2023-07-20

**Soundness:** 3

**Excitement:**

3: Ambivalent: It has merits (e.g., it reports state-of-the-art results, the idea is nice), but there are key weaknesses (e.g., it describes incremental work), and it can significantly benefit from another round of revision. However, I won't object to accepting it if my co-reviewers champion it.

**Paper Topic And Main Contributions:**

This paper introduces active learning and combines it with PET methods to further improve fune-tuning performance of NLU tasks in low-resource settings.

**Reasons To Accept:**

(1) This paper introduces active learning into PET field in  low-resource settings, which is ignored by previous research.

(2) The reprensentation is good with organzied sections and clear illustration by figures.


**Reasons To Reject:**

(1) The novelty that simply combines active learning methods with PET methods is a little bit incremental and sounds not that exciting.

(2) Validation on only four text classification datasets with on single model architecture (bert, encoder-only transformer) may cause  conclusion from the paper overclaim.


**Reproducibility:**

4: Could mostly reproduce the results, but there may be some variation because of sample variance or minor variations in their interpretation of the protocol or method.

**Reviewer Confidence:**

3: Pretty sure, but there's a chance I missed something. Although I have a good feel for this area in general, I did not carefully check the paper's details, e.g., the math, experimental design, or novelty.

---

> ### Author Rebuttal · Authors · 2023-08-28
>
> We thank the reviewer for the comments.
>
> > Validation on only four text classification datasets with on single model architecture (bert, encoder-only transformer) may cause conclusion from the paper overclaim.
>
> We appreciate the reviewer's feedback and concerns regarding potential overclaims. However, we would be grateful if the reviewer could specify which particular claim they found to be an overreach. In our paper, we have been careful to set the scope of our findings to the PLMs, AL strategies, and PEFT methods we investigated. We highlighted that "our results are specific to the PLMs, AL strategies, and PEFT methods examined in this study." We understand and acknowledge that – as for any other experimental paper submitted to ACL conferences – the results might vary when different models, strategies, or methods are applied. While we see the merit in testing a broader range of PLMs and datasets, computational constraints limited our choices, especially given the diversity of PEFT and AL methods we evaluated.

---

### Official Review · Reviewer_iSq3 · 2023-08-01

**Typos Grammar Style And Presentation Improvements:** 1. More training details, such as bat…
**Soundness:** 3

**Excitement:**

2: Mediocre: This paper makes marginal contributions (vs non-contemporaneous work), so I would rather not see it in the conference.

**Paper Topic And Main Contributions:**

The paper attempts to combine active learning and parameter-efficient tuning for large-scale pre-trained language models. The authors implement four active learning algorithms combined with four commonly used PEFT algorithms. Through experiments on four text classification datasets, they demonstrate that compared to full-parameter fine-tuning, PEFT is more effective in few-shot learning, and active learning further improves the performance of few-shot learning. Additionally, the authors conduct model analysis from the perspectives of forgetting dynamics and instance representations.

**Reasons To Accept:**

1. The paper explores the application of parameter-efficient tuning and task-adaptive pre-training in active learning and conducts extensive experiments to demonstrate the effectiveness of PEFT in active learning.
2. The authors analyze the reasons behind the better performance of PEFT from the perspectives of forgetting dynamics and instance representations, which can provide valuable insights for future work.

**Reasons To Reject:**

1. As mentioned in related work, previous studies have shown the importance of TAPT for active learning. However, the paper lacks a sufficient analysis of the challenges and specificity of combining PEFT and TAPT in the active learning framework, making the proposed method seem like a simple combination of two techniques.
2. Prompt tuning is also a common method in PEFT and has been proven to be effective in few-shot learning. It would be more convincing if the authors could provide relevant experiments on prompt tuning.
3. The current experiments focus on datasets with a limited number of sample categories. To better validate the effectiveness of the proposed method, it is recommended to select benchmark datasets with tens of categories, such as relation classification, etc.
4. The paper uses BERT as the base backbone. Existing work has demonstrated the good performance of large language models with over billions of parameters in few-shot learning. Including experiments based on LLMs will further strengthen the paper's persuasiveness.


**Reproducibility:**

3: Could reproduce the results with some difficulty. The settings of parameters are underspecified or subjectively determined; the training/evaluation data are not widely available.

**Reviewer Confidence:**

2: Willing to defend my evaluation, but it is fairly likely that I missed some details, didn't understand some central points, or can't be sure about the novelty of the work.

---

> ### Author Rebuttal · Authors · 2023-08-28
>
> We thank the reviewer for the valuable feedback.
>
> > As mentioned in related work, previous studies have shown the importance of TAPT for active learning. However, the paper lacks a sufficient analysis of the challenges and specificity of combining PEFT and TAPT in the active learning framework, making the proposed method seem like a simple combination of two techniques.
>
> We posit that integrating PEFT with TAPT can be straightforwardly achieved by pre-training the model on a masked language modeling task specific to a given dataset, with updates confined to the adapter parameters. Subsequent to this step, the active learning process is carried out as it would be with any other model. To our knowledge, this procedure doesn't introduce any unique challenges.
>
> > Prompt tuning is also a common method in PEFT and has been proven to be effective in few-shot learning. It would be more convincing if the authors could provide relevant experiments on prompt tuning.
>
> In our exploration of PEFT methodologies, we aimed to cover a diverse range of approaches. We did not prioritize prompt-tuning, mainly because studies [1,2] have demonstrated that soft prompt-tuning tends to underperform for smaller language models like BERT base, which we utilized in our research. This method appears to be more effective for considerably larger models, such as BERT large and RoBERTa large. Opting for a representative from the sequence optimization techniques, we chose prefix-tuning due to its consistent performance across varying sizes of language models [3].
>
> > The current experiments focus on datasets with a limited number of sample categories. To better validate the effectiveness of the proposed method, it is recommended to select benchmark datasets with tens of categories, such as relation classification, etc.
>
> We chose datasets that are frequently used in active learning research [4,5]. Among the multi-class datasets, we utilized TREC, which has six classes, and AG news with four classes. While we recognize the value of incorporating datasets with a greater number of classes, we were constrained by the available compute resources.
>
> > The paper uses BERT as the base backbone. Existing work has demonstrated the good performance of large language models with over billions of parameters in few-shot learning. Including experiments based on LLMs will further strengthen the paper's persuasiveness.
>
> We agree that the inclusion of larger language models would be advantageous. However, due to our hardware constraints, we opted for models that were within our computational capacity.
>
> ***
> We accidentally omitted the information regarding the batch size for the re-training step in each AL iteration and our use of default parameters for PEFT methods. We thank the reviewer for pointing it out. These details will be addressed in our updated version.
> Regarding learning rate optimization, it's customary in active learning to employ a consistent learning rate that is effective across various tasks [4,5], especially given the frequent absence of a validation set in real-world active learning contexts. In addition, we use early stopping that does not require a validation set.
>
> [1] What Changes Can Large-scale Language Models Bring? Intensive Study on HyperCLOVA: Billions-scale Korean Generative Pretrained Transformers (Kim et al., EMNLP 2021)\
> [2] The Power of Scale for Parameter-Efficient Prompt Tuning (Lester et al., EMNLP 2021)\
> [3] UniPELT: A Unified Framework for Parameter-Efficient Language Model Tuning (Mao et al., ACL 2022)\
> [4] Active Learning for BERT: An Empirical Study (Ein-Dor et al., EMNLP 2020)\
> [5] Revisiting Uncertainty-based Query Strategies for Active Learning with Transformers (Schröder et al., Findings 2022)

---

### Official Review · Reviewer_bhGY · 2023-08-04

**Soundness:** 3

**Excitement:**

3: Ambivalent: It has merits (e.g., it reports state-of-the-art results, the idea is nice), but there are key weaknesses (e.g., it describes incremental work), and it can significantly benefit from another round of revision. However, I won't object to accepting it if my co-reviewers champion it.

**Paper Topic And Main Contributions:**

The paper presents an empirical study on the behavior of parameter-efficient fine-tuning (PEFT) in low-resource settings for text classification tasks, specifically in the context of active learning (AL). The authors compare PEFT with full-fine tuning (FFT) and demonstrate the superiority of PEFT in low-resource settings, even in AL setups. They also analyze the forgetting dynamics and instance-level representations of PEFT and FFT, finding that PEFT yields more stable representations in early and middle layers. The paper highlights the synergistic potential of AL and PEFT in low-resource settings, emphasizing the advancements in efficient and effective fine-tuning.


**Questions For The Authors:**

1. My suggestion is to focus the article more on AL, and the PEFT section as a well defined related work would be fine.
2. More experiments on various tasks are still needed, or could you please explain why only text classification tasks were selected and how AL performs differently on this task compared to other tasks.
3. Can you provide original numbers of each method (i.e. accuracy). This will make the results appear more intuitive. Is there a significant improvement by combining AL into PEFT?
4. If AL affects the results, will the selection of 1000 data also affect the results? Have you tried sampling different data each time？

**Reasons To Accept:**

1. The paper explores the interplay between active learning and adapter-based parameter-efficient fine-tuning.
2. The authors provide empirical evidence to support their claims, conducting experiments on text classification tasks in low-resource settings.
3. The analysis of forgetting dynamics and instance-level representations further enhances the understanding of the behavior of these techniques.
4. The paper highlights the potential synergy between AL and PEFT, which can lead to advancements in efficient and effective fine-tuning.

**Reasons To Reject:**

1. Comparsion of PEFT and FFT lacks innovation. The experimental setup is no much different from previous work, and the authors only reprouduce the results on these datasets. Also, the viewpoints in the analysis （Sec 4.1）have also been discussed many times in previous work (e.g. Sec 4.3 in UNIPELT).
2. The paper focuses solely on text classification tasks in low-resource settings. Expanding the scope to include other domains or tasks would enhance the generalizability of the findings and make the research more applicable to a wider range of scenarios. Although the author mentioned this in the limitation, the comparative work has already been done on more tasks (e.g. GLUE Benchmark).
3. Table 2 is not intuitive enough. RIPL is already a relative indicator compared to before. The Table reports the relative results of the relative indicators, and it is difficult to see how much absolute improvement there has been.

**Reproducibility:**

4: Could mostly reproduce the results, but there may be some variation because of sample variance or minor variations in their interpretation of the protocol or method.

**Reviewer Confidence:**

3: Pretty sure, but there's a chance I missed something. Although I have a good feel for this area in general, I did not carefully check the paper's details, e.g., the math, experimental design, or novelty.

---

> ### Author Rebuttal · Authors · 2023-08-28
>
> We thank the reviewer for the comments and for raising a number of important questions.
>
> > Comparsion of PEFT and FFT lacks innovation. The experimental setup is no much different from previous work, and the authors only reprouduce the results on these datasets.
>
> Please note that our primary interest lies in the intersection of AL and PEFT in **low-resource** settings. This is an aspect we believe hasn't been explored previously, which we also highlight in the paper: “However, these findings [of previous research] were based on comparing FFT to a single adapter variant on a full dataset or evaluating the performance at only a few discrete points.” Thus, even though PEFT's efficacy is well-documented, we offered detailed learning curves in low-resource settings. However, for reason of completeness and as a sanity check, we did undertake an initial comparison between PEFT and FFT.
>
> > RIPL is already a relative indicator compared to before. The Table reports the relative results of the relative indicators, and it is difficult to see how much absolute improvement there has been.
>
> We acknowledge that the RIPL metric reports the proportion of relative improvement, which makes it inappropriate to measure absolute improvement. Nevertheless, we believe that the RIPL metric allows for a clear comparison of AL methods across different datasets, given that it measures the relative improvement over random sampling. In addition, to provide a comprehensive perspective, we present active learning curves that illustrate the absolute performance scores, as depicted in Figure 3. We believe that offering both the AL learning curves and the relative improvement contributes to a more comprehensive understanding of the results.
>
> > Q1: …why only text classification tasks were selected and how AL performs differently on this task compared to other tasks
>
> Our emphasis is on text classification tasks, given their prominent coverage in AL research. As we explore novel combinations like PEFT with AL, we wanted to make sure that our results are comparable with existent AL literature [1,2]. Regarding the performance on other types of tasks such as sequence labeling, AL has also shown improvement over random sampling over various domains (e.g., [3,4]). We certainly agree that including additional tasks would be valuable, but we decided to stick to classification due to resource limitations.
>
> > Q2: Can you provide original numbers of each method (i.e. accuracy). This will make the results appear more intuitive. Is there a significant improvement by combining AL into PEFT?
>
> If we have understood correctly, the question is whether we can provide performance scores in isolation. We have indeed reported the performance in terms of the F1 score for each PEFT method in isolation and in combination with every AL method, comparing it to FFT. Additionally, we included the results with and without applied TAPT, which are all shown as learning curves in Figure 3. For each setting, we show the learning curve with F1 scores at each point, corresponding to specific numbers of available training instances ranging from 100 to 1000. As reported in the paper, AL shows improvement over random sampling across the board (on datasets we used) when combined with PEFT.
>
> > Q3: If AL affects the results, will the selection of 1000 data also affect the results? Have you tried sampling different data each time?
>
> We presume that the question pertains to the effect of retrieving different random samples, i.e., having multiple runs. To account for that, we repeat each experiment five times with different random seeds as stated in the paper: “For every setting, we perform five runs using different random seeds. We report the average F1 score at each sampling step (with and without AL for FFT and PEFT) to show the corresponding learning curve averaged over five runs.” We are effectively drawing different random samples in each AL step for different runs.
>
> [1] On the Importance of Effectively Adapting Pretrained Language Models for Active Learning (Margatina et al., ACL 2022)\
> [2] Active Learning for BERT: An Empirical Study (Ein-Dor et al., EMNLP 2020)\
> [3] Active Learning for Sequence Tagging with Deep Pre-trained Models and Bayesian Uncertainty Estimates (Shelmanov et al., EACL 2021)\
> [4] SeqMix: Augmenting Active Sequence Labeling via Sequence Mixup (Zhang et al., EMNLP 2020)

---

### Meta-Review · Area_Chair_dwCm · 2023-09-19

**Recommendation:** 4

**Metareview:**

This work explores parameter-efficient fine-tuning (PEFT) in the context of active learning (AL). The authors show that PEFT methods outperform full fine-tuning in low-resource conditions, and the performance gain with PEFT methods extends to AL settings.

1. As the reviewer mentioned, the authors present a large set of experiments that includes 4 PEFT methods, 5 AL sampling strategies, and 4 classification datasets. Experimental results are sound and support the authors' claims.

2. Analysis of forgetting dynamics gives insights into why PEFT methods perform differently in AL scenarios.

Furthermore, the authors provide clarification regarding their experimental setup. Overall, the benefits of PEFTs in AL is an important angle, and the paper addresses such benefits. I believe the authors will take reviewers' suggestions and points into account, such as providing the absolute performance metrics of PEFTs and FFT in AL settings similar to Table 1.

---

### Decision · Program_Chairs · 2023-10-07

**Decision:**

Accept-Main

**Comment:**

This work explores parameter-efficient fine-tuning (PEFT) in the context of active learning (AL). The authors show that PEFT methods outperform full fine-tuning in low-resource conditions, and the performance gain with PEFT methods extends to AL settings.

1. As the reviewer mentioned, the authors present a large set of experiments that includes 4 PEFT methods, 5 AL sampling strategies, and 4 classification datasets. Experimental results are sound and support the authors' claims.

2. Analysis of forgetting dynamics gives insights into why PEFT methods perform differently in AL scenarios.

Furthermore, the authors provide clarification regarding their experimental setup. Overall, the benefits of PEFTs in AL is an important angle, and the paper addresses such benefits. I believe the authors will take reviewers' suggestions and points into account, such as providing the absolute performance metrics of PEFTs and FFT in AL settings similar to Table 1.